# Monitoring of Pseudorabies in Wild Boar of Germany—A Spatiotemporal Analysis

**DOI:** 10.3390/pathogens9040276

**Published:** 2020-04-10

**Authors:** Nicolai Denzin, Franz J. Conraths, Thomas C. Mettenleiter, Conrad M. Freuling, Thomas Müller

**Affiliations:** 1Friedrich-Loeffler-Institut, Institute of Epidemiology, 17493 Greifswald-Insel Riems, Germany; Nicolai.Denzin@fli.de (N.D.); Franz.Conraths@fli.de (F.J.C.); 2Friedrich-Loeffler-Institut, 17493 Greifswald-Insel Riems, Germany; Thomas.Mettenleiter@fli.de; 3Friedrich-Loeffler-Institut, Institute of Molecular Virology and Cell Biology, 17493 Greifswald-Insel Riems, Germany; Conrad.Freuling@fli.de

**Keywords:** Germany, monitoring, pseudorabies, Aujeszky’s disease, spatiotemporal analysis, relative risk, wild boar

## Abstract

To evaluate recent developments regarding the epidemiological situation of pseudorabies virus (PRV) infections in wild boar populations in Germany, nationwide serological monitoring was conducted between 2010 and 2015. During this period, a total of 108,748 sera from wild boars were tested for the presence of PRV-specific antibodies using commercial enzyme-linked immunosorbent assays. The overall PRV seroprevalence was estimated at 12.09% for Germany. A significant increase in seroprevalence was observed in recent years indicating both a further spatial spread and strong disease dynamics. For spatiotemporal analysis, data from 1985 to 2009 from previous studies were incorporated. The analysis revealed that PRV infections in wild boar were endemic in all German federal states; the affected area covers at least 48.5% of the German territory. There were marked differences in seroprevalence at district levels as well as in the relative risk (RR) of infection of wild boar throughout Germany. We identified several smaller clusters and one large region, where the RR was two to four times higher as compared to the remaining areas under investigation. Based on the present monitoring intensity and outcome, we provide recommendations with respect to future monitoring efforts concerning PRV infections in wild boar in Germany.

## 1. Introduction

Pseudorabies virus (PRV) is an enveloped double-stranded DNA virus that causes Aujeszky’s disease (AD), which affects swine and other mammals [1]. Taxonomically, PRV is synonymous with suid herpesvirus 1 (SuHV-1), a member of the genus *Varicellovirus*, subfamily Alphaherpesvirinae of the Herpesviridae family [2]. Despite considerable advances in controlling and eliminating the disease in domestic pig populations in Europe, North America, Australia and New Zealand in the recent past [1,3,4,5], there has been increasing evidence for the widespread occurrence of PRV in populations of free-roaming and farmed wild swine—an umbrella term for both true wild boar and feral swine as well as hybrids thereof—in many regions of its range [6,7,8]. In Germany, wild boar are abundant across the entire country with varying population densities, based on wild boar hunting statistics (Figure 1).

During the last four decades, intensive research has improved our understanding of the role wild swine populations play in the epidemiology of PRV. Serosurveys have confirmed the occurrence of PRV infection in populations of wild boar and feral swine in at least 16 European countries and the US. These data indicate a large-scale but patchy distribution pattern with seroprevalences ranging from 0% to 72% depending on the region [7,8,9]. PRV-specific antibodies have also been detected in wild boar in coastal ranges of Northern Africa [10], northern Turkey [11], Japan [12,13] and the Korean peninsula [14]. In contrast, PRV infections appeared to be absent in wild swine of New Zealand, probably due to geographical isolation [15]. However, occasional PRV isolations from free-roaming wild swine or hunting dogs suggest a far more widespread circulation of PRV in populations of wild boar than previously assumed, even in countries where a serological monitoring routine has not yet been implemented [16].

PRV infections in wild swine appear to be caused by highly adapted variants of PRV [3,16]. Transmission among wild swine mainly occurs via the venereal [17,18] and oronasal routes [3] and to a lesser extent through cannibalism, with the latter restricted mainly to feral swine in the US [19]. Phylogenetic analysis based on partial sequencing of the 5k non-coding and coding regions of the gene encoding glycoprotein C suggests that PRV isolates of wild boar origin do not represent a homogenous virus population. While there is evidence for a circulation of several genetic lineages of PRV strains in populations of wild swine in the US and Europe, the high genetic diversity indicates multiple introductions from domestic pigs into their wild living conspecifics in recent history [3,16,20]. Due to the latent character of the infection, elimination of PRV in populations of wild swine at a local or regional level is considered difficult if not impossible [7]. 

As in many other European countries, PRV is also present in the wild boar population in Germany, where infections in wild boar are known to be endemic, particularly in the eastern parts of the country (the former German Democratic Republic), a region that comprises about one-third of Germany’s total area [21,22,23,24]. Continuous serological monitoring of wild boar in this particular area over more than two decades (1985–2008) has provided valuable insights into the evolution, spread and dynamics of wild boar-mediated PRV infections [25,26]. PRV seroprevalences were estimated at 16.5% on average for the whole region but reached up to 45% in endemic hot spots [25]. In contrast, serosurveys in western and southern parts of Germany were rather fragmentary and often spatially and temporally limited [27,28,29,30]. These factors have hampered obtaining a sound assessment of the spread of PRV infections in wild boar at a national level.

In order to overcome the latter limitations, a nationwide serological monitoring was initiated in Germany in 2010. The main objectives of this study were to: (i) gain an overview of the dimension of the spread of PRV infections in wild boar at a national level, (ii) estimate PRV seroprevalences at a regional level, and (iii) identify high-risk areas by determining the spatial relative risk (RR) and (iv) revise recommendations for effective future monitoring of PRV infection in wild boar based on the obtained results.

## 2. Materials and Methods 

### 2.1. Study Area and Sample Collection

The study area encompassed the entire territory of Germany, covering a total area of 357,386 km² and included 14 major territorial federal states but excluded the City States of Hamburg (HH) and Bremen (HB; Table 1, Figure 2). Between October 2010 and January 2016, wild boar blood samples were obtained and submitted to the responsible regional veterinary laboratories as previously described [25]. Data were recorded for each sample, including sex, age, date and geographical location of origin. Sample submissions were recorded per hunting season (October until January of the following calendar year).

### 2.2. Diagnostic Assays

Upon arrival in the laboratory, blood samples were centrifuged at 1,000 *g* for 10 min, the serum was recovered, aliquoted, labelled with a unique barcode and subsequently stored at –20 °C prior to testing. The samples were tested for the presence of PRV-specific antibodies using six different commercial PRV-glycoprotein B (gB)- or PRV-glycoprotein E (gE)-based enzyme-linked immunosorbent assays (ELISAs) licensed by the Friedrich-Loeffler-Institut (FLI) pursuant to section 11 of the German Animal Health Act. The utilised antibody ELISA tests included the SVANOVIR PRV gB-Ab/gE-Ab (Boehringer Ingelheim Svanova), the IDEXX PRV/ADV gI, IDEXX PRV/ADV gB (IDEXX Europe B.V.), the ID Screen Aujeszky gB Competition (ID VET), the PrioCHECK PRV gB (Thermo Fisher Scientific) and the SERELISA Aujeszky gI N assay (Zoetis France). Testing of the sera strictly followed the manufacturers’ instructions. 

### 2.3. Spatiotemporal Analysis

A descriptive spatiotemporal analysis was performed based on the results of the concerted nationwide PRV monitoring (2010–2015). For a more detailed analysis, this data set was combined with data from previous surveys conducted in six federal states of Eastern Germany between 2000 and 2009 [25], covering a total observation period of 16 years. Spatial analysis comprised the calculation of a relative risk (RR) surface with cluster detection depending on point data. Since no exact geo-coordinates were available for the sampled wild boar, locations were allocated to the centroids of the smallest administrative units, i.e., city/village or municipality/district. Additionally, the data were evaluated as aggregated in administrative units. To assess potential temporal and spatiotemporal differences in PRV seroprevalence, the combined data set for the entire observation period (2000–2015) was subdivided into two time intervals using the median of the submission date of the samples as a threshold. RR surfaces, seroprevalences in administrative units and overall PRV seroprevalence estimates with 95% confidence interval limits calculated according to the Clopper–Pearson method [31] were determined for the entire observation period (2000–2015) and separately for the two time intervals. In order to assess the dimension and direction of a potential spatial selection bias, seroprevalence estimates were adjusted for the geographic origin of the samples as previously described [32]. Finally, the probability of presence (endemicity) or absence of PRV infections in wild boar in Germany was evaluated at the district level.

#### 2.3.1. Relative Risk

The approximated RR of a wild boar within Germany to test positive for PRV-specific antibodies was calculated separately for the entire pbservation period and for the two time intervals; these data were illustrated using the R package ‘sparr’ as previously described [33]. Using this method, the kernel density estimations [34] (Gaussian kernel, bandwidth chosen as fix) of the cases (ELISA-positive wild boar), as well as of the overall samples (ELISA-positive and -negative wild boar, basic data set), were calculated separately for a grid with a cell resolution of 1000 m × 1000 m in Germany. The ratios of the integrals of the standardised kernel densities of the cases and all samples (in each grid cell) were used to illustrate the function of RR [35,36]. The bandwidth of the kernel density estimations of 13.4 km was determined using the mean integrated squared error [37]. It was used for the interpolation of cases (ELISA-positives) and all sample data (ELISA-positives and -negatives) [38]. Edge correction was performed as described elsewhere [39]. Regions with a statistically significant increase in RR were detected and highlighted by calculating *p*-value contour lines [40,41]. All calculations were performed in R 3.6.0 [42].

#### 2.3.2. Seroprevalence in Administrative Units

Spatiotemporal dynamics in PRV seroprevalence in wild boar were displayed for the different time intervals using choropleth maps of Germany at district level. To account for uncertainties related to cumulative datasets at the district level [43], empirical Bayes estimation [44] was applied prior to mapping and to calculate the empirical Bayes prevalence. The latter is a weighted mean of the regional raw prevalence and the global prevalence [45]. The weighting was based on the variability of the estimates, whereby raw estimates were adjusted towards the global PRV seroprevalence estimate to reduce bias in survey estimates for small sample sizes at district level [45].

#### 2.3.3. Assessment of the Potential Effect of a Spatial Selection Bias

To account for the potential influence of spatial variation of the RR of a wild boar to test positive for PRV-specific antibodies within the study area, the PRV seroprevalence estimates for the two time intervals were adjusted according to the origins of the underlying samples and their RR to test seropositive or -negative as previously described [32]. The spatiotemporal RR surface for the study area was calculated as described below.

The conditional probability (p) of a new ‘seropositive’ sample at a certain geographic origin y is:(1)$$p=\frac{f\left(y\right)}{g\left(y\right)+f\left(y\right)},$$
where f(y) denotes the kernel density estimate of the cases and g(y) of the samples testing negative, respectively [36]. The RR at the geographic origin y is then expressed as the ratio of p and the overall prevalence in the territory. In order to account for spatially unrepresentative sampling when comparing prevalences, the RR can be used to weight positive and negative sample results according to their origin. Positive results from high-risk areas were weighted down, negative ones up and vice versa in low-risk areas. To exclude the effect of a sample result on its own weight, the density estimates were reduced by the maximum density value produced by a single sample and n was reduced by one in the calculations. The weights used for positive and negative samples, respectively, were therefore calculated as follows:(2)$$\omega_{pos}=\frac{\left(n_{pos}-1\right)\left(g\left(y\right)+f\left(y\right)-Z_{max}\right)}{\left(f\left(y\right)-Z_{max}\right)\left(n_{con}-1\right)}$$
(3)$$\omega_{neg}=\frac{\left(n_{neg}-1\right)\left(g\left(y\right)+f\left(y\right)-Z_{max}\right)}{\left(g\left(y\right)-Z_{max}\right)\left(n_{con}-1\right)},$$
where Z_max_ denotes the maximum density value of one event and n_pos_ and n_con_ denotes the number of positive samples and controls (positive and negative samples) in the area of Germany. Kernel density estimates were calculated by employing the R package ‘sparr’ [33].

#### 2.3.4. Evaluation of the Probability of Disease Presence

The probability of disease presence (endemicity) at the district level was expressed by applying a 5% threshold. In districts with negative samples only, the probability (P) of not detecting the disease (at least one positive specimen in the sample) at a design prevalence of 5% (assumed prevalence if the disease was present) was calculated as a function of the size of the tested sample: (4)$$P={\left(1-p_{\mathrm{des}}\right)}^{n},$$
where P_des_ denotes the design prevalence and n the sample size in a district.

#### 2.3.5. Hunting Statistics

Data on the average wild boar hunting bag for the years 2014/15–2017/18 (Figure 1) were obtained from the database ‘Datenspeicher Jagd’ of the Thünen-Institute for Forest Ecosystems, Federal Research Institute for Rural Areas, Forestry and Fisheries, Eberswalde, Germany in form of a map and made available through the German Hunting Association (Deutscher Jagdverband—DJV; map basis: GeoBasis-DE/BKG 2017). 

## 3. Results

### 3.1. Nationwide PRV Monitoring in Wild Boar (2010–2015)

Between October 2010 and March 2016—a period that represents six hunting seasons—a total of 108,748 wild boar blood samples were collected in the 14 major territorial federal states of Germany. Sampling intensity varied considerably at the regional and district level during the observation period. Eleven federal states conducted serosurveys during at least four of the six years’ observation period, while two, Berlin (BE) and Mecklenburg-Western Pomerania (MWP), only tested samples in one year (Table 1, Figure 2). The majority of samples (73.9%) originated from the federal states of Saxony (SN), Hesse (HE), Bavaria (BY) and Lower Saxony (LS). Approximately 57% of the serum samples were obtained during the years 2013 to 2015. 

Of the 108,748 serum samples tested by ELISA, a total of 15,311 (14.1%) were positive for PRV-specific antibodies (Table 1). The average periodic PRV seroprevalences for the observation period ranged from 0.14% in Schleswig Holstein (SH) to 31.4% in SN. In five federal states, BY, BE, MWP, SN and Saxony-Anhalt (ST), the average periodic PRV seroprevalence was greater than 10%. In contrast, the average periodic PRV seroprevalences in the remaining federal states were much lower (Table 1).

### 3.2. Spatiotemporal Analysis of the Combined Data Set (2000–2015)

The combined data set for the entire observation period 2000–2015 used for the spatiotemporal analysis comprised a total of 149,594 individual wild boar serum samples that were tested for the presence of PRV-specific antibodies. The allocation of test results and the accuracy of available information concerning the geographic origin of the samples is shown in Table 2.

Using the median of the submission date of samples as a threshold, two time intervals were defined: time interval 1 and 2 comprised the years 2000–2012 and 2012–2015, respectively. Given that information on the time point of sampling was limited to the year of submission and there were 18,042 samples tested in the median year 2012, 5692 samples from 2012 were randomly selected and allocated to interval 1 and the remaining samples to interval 2, in order to assure an equivalent sample size in both time intervals (74,797; Table 3).

The nationwide overall PRV seroprevalence in wild boar for the entire observation period (2000–2015) was estimated at 13.7%. The prevalence increased slightly, but significantly from time interval 1 (2000–2012, 12.9%) to time interval 2 (2012–2015, 14.5%). Details are provided in Table 3. Prevalence estimates adjusted for the origin of the samples confirmed the increase in prevalence. These estimates are not meant to replace the original estimates; rather, they provide some insight into the potential dimension and direction of a spatial selection bias. The adjusted estimates are therefore provided without confidence intervals in Table 3.

#### 3.2.1. Spatiotemporal Analysis of the RR

A spatiotemporal analysis of the RR revealed that the approximated RR, i.e. the probability of a wild boar in a given area testing positive for PRV-specific antibodies relative to the average probability in the entire area of Germany, was between zero and four for the entire observation period and for the two time intervals. At least eight spatial clusters—regions with a significantly elevated RR (*p* > 0.05)—were identified in varying sizes throughout the study area. Within these clusters, the probability of a wild boar testing positive for PRV-specific antibodies was two to four times higher compared to the average probability of the remaining areas under investigation (Figure 3). While most of the clusters were relatively small, there was one large spatial cluster identified in the Eastern part of Germany irrespective of the investigated time interval, in which the RR was particularly high. This high-risk area seemed to shrink over time, with hotspots in the very North-East of the country disappearing from time interval 1 (2000–2012) to time interval 2 (2012–2015), while the southern part of the large cluster remained quite stable over time (Figure 3).

#### 3.2.2. Spatiotemporal Analysis of the PRV Seroprevalence at District Level

Serological data were available for 338 (81.8%) of the 413 districts of Germany. PRV seropositive animals were detected in 144 (34.9%) of the districts, covering 48.5% of Germany’s territory. Raw estimates and empirical Bayes estimates of PRV seroprevalences at the district level ranged from 0%–66.7% (median: 0%; mean: 5.4%) and 0.06%–37.6% (median: 4.0%; mean: 7.3%) for the entire observation period (2000–2015; Figure 4), respectively. When broken down by time interval, the raw and empirical Bayes estimates for time interval 1 (263 districts with data) were 0%–60.0% (median: 0%; mean 5.5%) and 0.08%–40.0 % (median: 5.7%; mean 8.3%), respectively. The estimates (raw and Bayes) for time interval 2 (306 districts with data) differed only slightly from time interval 1, namely 0%–66.7% (median: 0%; mean: 4.8%) and 0.1%–39.8 % (median: 4.0%; mean: 7.0%), respectively.

Estimated PRV seroprevalences were highest in the Eastern part of Germany. Here, during time interval 2 (2012–2015), the PRV seroprevalence aggregated for the region had increased to 25.6%. Particularly in the Southeastern parts of this area, there were districts that formed a large contiguous area comprising parts of the federal states of MWP, BB, BE and SN, where PRV seroprevalences reached values of 20% and higher. However, there were also single districts in the Central, Western and Southern parts of the country with similarly high PRV seroprevalences. This pattern did not change considerably between the two time intervals (Figure 3 and Figure 4).

#### 3.2.3. The Probability of Endemicity and Absence of Disease

Considering the entire observation period (2000–2015), there were 75 districts without data (18.2% of the districts, 6.1% of the area of Germany). For these districts, PRV endemicity in wild boar cannot be ruled out, since there was no testing. Concerning districts with available data and at least one positive test result, raw seroprevalences were below 5% for 55 districts (13.3% of all districts) and greater than or equal to 5% for 89 districts (21.6% of all districts). In the latter districts with positive test results, PRV in wild boar may be considered as endemic with a high or very high probability, respectively, if a 5% threshold was applied (Figure 5). For districts that submitted samples yielding negative test results only (53.0% of all districts), the probability of not detecting the disease if present at a 5% design prevalence is displayed in Figure 5. For 83 of those aforementioned districts (20.1% of all districts), the probability of a false freedom assumption is less than 1%.

## 4. Discussion

PRV infections in wild boar are endemic throughout Europe and the southern parts of the US [7,8,9]. The overabundance of wild boar in many parts of its range represents a risk factor for the transmission of PRV as well as other pathogens of domestic animals and humans [46]. According to chapter 8.2 of the OIE Terrestrial Code, appropriate measures must be implemented to prevent any transmission of PRV from wild boar and feral pigs to domestic and captive wild pigs [47]. A prerequisite for the implementation of such measures would be exact knowledge about the presence and the extent of the geographical spread of PRV infections in wild boar . However, assessing the full extent and spatial spread of PRV in wild boar is difficult. Part of the problem is that there is no legal obligation at the national and international level to monitor PRV infections in wild boar. This endeavour is often perceived as a financial burden; hence, most serosurveys conducted in Europe in recent years have been limited in time and scope (for a review, see [7]). In fact, only a very few countries have implemented monitoring campaigns at a national level [48,49,50,51,52,53].

While previous approaches to assess PRV seroprevalence in wild boar populations of Germany mainly focused on East Germany [22,24,25,26] and only occasionally targeted regions in other parts of the country [21,27,28,29], this study represents the first nationwide monitoring of PRV infection in wild boars. Unfortunately, for reasons mentioned above, the monitoring was spatially and temporally incomplete (Table 1, Figure 2). Despite the voluntary nature of participation in the nationwide monitoring, the idea spread over time, and the ranks of federal states were closed according to the snowball effect, so that almost the entire territory of Germany was eventually covered (Figure 2). With more than 108,000 serum samples tested over a six-year observation period (Table 1), this is by far the most comprehensive study conducted so far, topping even the sample size for Eastern Germany obtained during a 14-year observation period [25]. For comparison, sample sizes in Poland and France amounted to 27,263 and 11,533 [53,54], respectively, while six other studies tested between 2379 and 7021 sera for the presence of PRV specific antibodies in recent years [22,23,55,56,57,58].

The overall nationwide PRV seroprevalence of 12.09% in wild boar for the time period 2010 to 2015 (Table 1) resembles that reported for PrV in feral swine populations in the Czech Republic, France, Slovenia, Greece, Poland and the United States, where the overall seroprevalence has been estimated to range from 10% to 45% at the national level [8,48,49,50,51,52,59]. Previous evidence suggested the occurrence of PRV infections in several parts of Germany, including the federal states of LS, NW and East Germany [21,23,25,27,28,29]. Predictably, nationwide monitoring for the first time confirmed that PRV infections in wild boar are far more widespread in Germany—wild boar populations of all federal states are affected, albeit to different degrees (Table 1, Figure 4).

The combination of the data sets from the concerted nationwide PRV monitoring (2010–2015, Table 1) with data sets from a previous survey conducted in six federal states of Eastern Germany between 2000 and 2009 [25] (Table 2) provided high levels of precision in the estimation of the overall PRV seroprevalence (Table 3). They afforded the unique opportunity to gain more insights into the spatiotemporal dynamics of PRV infections during a 16 years’ observation period. 

The overall PRV seroprevalence estimates for the combined period almost equalled those of the nationwide monitoring conducted between 2010 and 2015 but showed a significant increase between the two time intervals (Table 3). To provide some insight into the potential dimension and direction of a spatial selection bias, the seroprevalence estimates were adjusted for the geographic origin of the samples, and the increase in PRV seroprevalence was confirmed (Table 3). The analysis of the combined data set revealed that Eastern Germany seemed to be a hotspot for PRV infections in wild boar because seroprevalences were particularly high in this area (Figure 3, Figure 4 and Figure 5). Nonetheless, the 14.5% overall PRV seroprevalence obtained for the entire territory of Germany in recent years (time interval 2, 2012–2015; Table 2) almost equalled the prevalence for the Eastern parts of Germany in 2008, when the PRV seroprevalence peaked at 15.9% after a long period of evolution [25]. Interestingly, if only data from Eastern Germany were analysed for the same time interval, the PRV seroprevalence in this particular region had increased by almost 10%.

Even for the Eastern parts of Germany, it has been suggested that the increase in PRV seroprevalence over a 24 year observation period (1985–2008) is a consequence of strong disease dynamics, which may have led to a westward spread at an average speed of 3.3 km/year [25,26,60]. However, phylogenetic evidence suggests at least two different PRV variants are circulating in wild boar populations in Germany. While clade A PRVs predominantly occur in the Eastern, Middle and Northern parts of Germany, clade B viruses are only found in the far West of the country [16], where they are part of large transboundary infectious cycles [7]. High PRV seroprevalences in wild boar found in neighbouring countries, including Poland, the Czech Republic and Belgium [48,53,61], support this hypothesis. Furthermore, adjacent PRV endemic areas most likely constitute common clusters with those found in border areas in Germany (Figure 4). Although these data provide only a snapshot, there is reason to believe that the current PRV situation in wild boar in Germany is the result of a continuous spread from various directions (Figure 3). 

This spatiotemporal risk surface analysis identified a few spatial clusters with an increased RR (Figure 3). Apart from the large cluster in Eastern Germany, there were rather isolated smaller, but persistent foci of infection, where the RR was two to four times higher compared to other areas under investigation. The observation that the high-risk area in Eastern Germany seemed to shrink during time interval 2 (Figure 3) is probably an artefact due to missing data from this area for the respective time period (Figure 4, time interval 2). Given that wild boar populations have generally increased—in both size and geographic distribution—across Europe over the past 30 years [62,63,64], one explanation could be high population densities of wild boar in these spatial clusters, as was found in a previous study [25]. It should again be emphasised that the RR as assessed in this study represents the ratio of the probability of a wild boar in a given area testing seropositive for PRV and does not provide any information about the absolute risk of infection. Therefore, the RR surface (Figure 3) only indicates PRV hotspot areas, where obviously the transmission of PRV within the wild boar population has reached an optimum. The real dimension of the spread of PRV in wild boar is wider (Figure 3).

## 5. Conclusions

Given the high population density of wild boar in Germany (Figure 1) [62], the self-contained circulation of highly adapted, low-virulent PRV strains [65,66] and the virtually endemic status of large parts of the country, it is just a matter of time until the entire territory of Germany will be affected by PRV in wild boar. Generally, monitoring PRV in wild boar makes sense, although previous evidence suggests that once an area is serologically determined as endemic for PRV, the infection will prevail and further surveillance provides hardly any additional information [67]. Recent data from the US [8] and the data reported here are in accord with this view. As a consequence, future monitoring activities of PRV in wild boar of Germany should focus exclusively on currently seronegative districts or districts without any data in the respective federal states, considering the probability of false freedom assumptions (Figure 5). 

Germany last faced an outbreak of Aujeszky’s disease in domestic pigs in 1999 and has been officially free from the disease in domestic pigs since 2003 [5]. Despite the massive occurrence of PRV infections in wild boar, no spillover into domestic pigs has been reported since that time, although this cannot be ruled out for the future. Generally, a sound biosecurity regime on farms should be maintained to avoid spillover of PRV and other pathogens.

## Figures and Tables

**Figure 1 pathogens-09-00276-f001:**
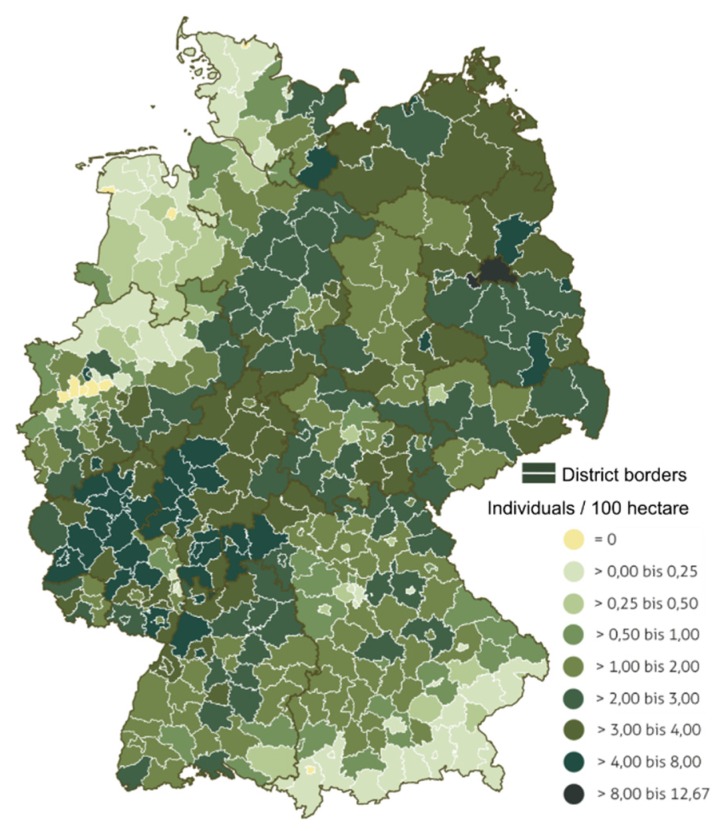
Average wild boar hunting bag for the years 2014/15–2017/18 (modified from data of the German Hunting Society, 2019).

**Figure 2 pathogens-09-00276-f002:**
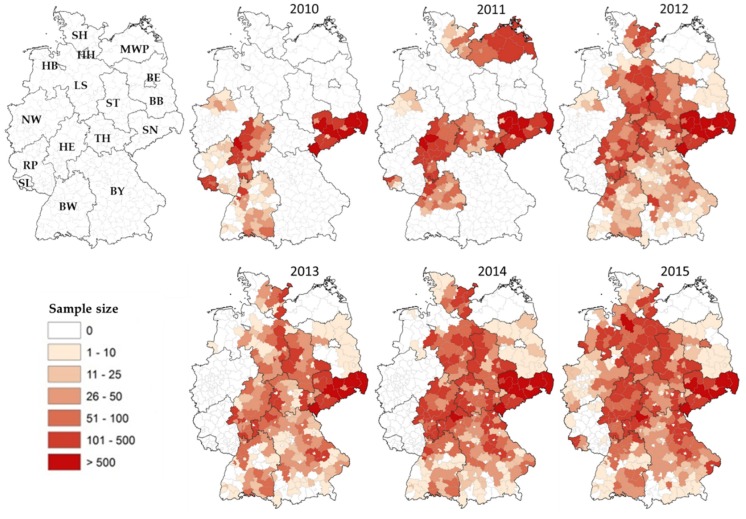
Implementation of serological PRV monitoring in wild boar and sample sizes obtained in the 16 German federal states from 2010 to 2015. Federal state abbreviations: BE—Berlin, BB—Brandenburg, BW—Baden-Württemberg, BY—Bavaria, HB—Bremen, HH—Hamburg, HE—Hesse, LS—Lower Saxony, MWP—Mecklenburg Western Pomerania, NW—North Rhine Westphalia, RP—Rhineland Palatinate, SH—Schleswig-Holstein, SL—Saarland, SN—Saxony, ST—Saxony-Anhalt, TH—Thuringia.

**Figure 3 pathogens-09-00276-f003:**
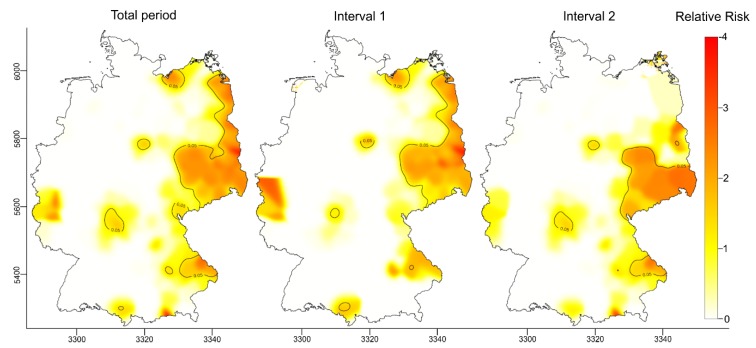
Relative risk (RR) of a positive PRV antibody test result in German wild boar from 2000 to 2015 framed by geographic coordinates (Universal Transverse Mercator (UTM), Zone 33, with coordinates divided by 1000). Values in RR above one indicate an elevated risk and below one a reduced risk relative to the average risk in the overall territory. Areas with a statistically significant increase in the RR at the 0.05 level are encircled by solid contour lines.

**Figure 4 pathogens-09-00276-f004:**
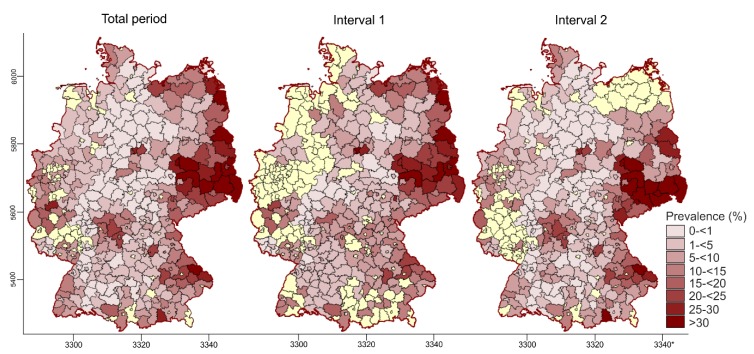
Weighted (empirical Bayes estimation) PRV seroprevalence in wild boar of Germany at the district level (2000–2015) framed by geographic coordinates (Universal Transverse Mercator (UTM), Zone 33, with coordinates divided by 1000).

**Figure 5 pathogens-09-00276-f005:**
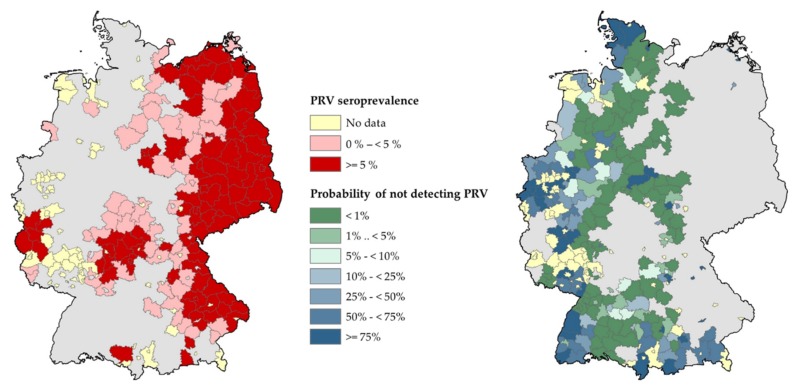
Map of Germany highlighting (A) the probability of PRV endemicity in populations of wild boar at the district level based on raw seroprevalences and (B) the probability of not detecting the PRV infections in thus far PRV seronegative districts (2000–2015).

**Table 1 pathogens-09-00276-t001:** Nationwide pseudorabies virus (PRV) monitoring in wild boar (2010–2015): number of wild boar sera submitted for testing on the presence of PRV-specific antibodies per German federal state and year.

Federal State	2010			2011			2012			2013			2014			2015			Total			
	N	pos	prev	N	pos	prev	N	pos	prev	N	pos	prev	N	pos	prev	N	pos	prev	N	pos	prev	95% CI
BE																91	14	15.38	91	14	**15.38**	**8.67–24.46**
BB							73	6	8.22	52	4	7.69	131	13	9.92	97	0	0	353	23	**6.52**	**4.17–9.62**
BW	768	18	2.34	1131	4	0.35	1118	15	1.34	1062	14	1.32	1212	22	1.82	1286	26	2.02	6577	99	**1.51**	**1.23–1.83**
BY							1575	81	5.14	2798	287	10.26	4606	550	11.94	4125	405	9.82	13,104	1323	**10.10**	**9.59–10.62**
HB																						
HE	2895	35	1.21	4900	64	1.31	3015	54	1.79	1751	33	1.88	2909	68	2.34	3073	72	2.34	18,543	326	**1.76**	**1.57–1.96**
HH																						
LS							2760	222	8.04	1305	56	4.29	1939	11	0.57	4430	22	0.50	10,434	311	**2.98**	**2.66–3.33**
MWV				1104	135	12.23													1104	135	**12.23**	**10.35–14.31**
NW	78	0	0	68	0	0	53	0	0							799	3	0.38	998	3	**0.30**	**0.06–0.88**
RP	100	5	5.00	,_	_	_	_	_	_	_	_	_	1936 **	47 **	2.43 **	9	1	11.11	2092	6	**0.29**	**0.11–0.62**
SH				279	2	0.72	848	0	0	619	0	0	1184	3	0.25	599	0	0	3529	5	**0.14**	**0.05–0.33**
SL	1397	2	0.14	330	1	0.30										373	25	6.70	2100	28	**1.33**	**0.89–1.92**
SN	5695	1624	28.52	5464	1665	30.47	6889	2048	29.73	6542	2171	33.19	7527	2585	34.34	6258	2.126	33.97	38,375	12,219	**31.84**	**31.38–32.31**
ST							996	79	7.93	936	122	13.03	1101	138	12.53	1855	182	9.81	4888	521	**10.66**	**9.81–11.56**
TH				917	18	1.96	1343	38	2.83	1028	41	3.99	1983	88	4.44	1336	66	4.94	6607	251	**3.80**	**3.35–4.29**
**Total**	**10,933**	**1684**	**15.40**	**14,193**	**1889**	**11.71**	**18,670**	**2543**	**13.62**	**16,093**	**2728**	**16.59**	**22,592**	**3478**	**15.39**	**24,331**	**2.942**	**12.09**	**108,748**	**15,311**	**14.08**	**13.87–14.29**

N = number of serum samples submitted; pos = number of test-positive samples (ELISA); prev = seroprevalence (percentage of test-positive samples); CI = confidence interval **cumulative data for the years 2011–2014; see Figure 2 caption regarding abbreviations for the federal states.

**Table 2 pathogens-09-00276-t002:** Allocation and accuracy of geographic coordinates of sample origin.

Administrative Unit	Town/Village	Municipality	District	Total
**No. of Cases ***	5274	380	14,898	20,552
**No. of Controls ^#^**	43,906	16,023	89,665	149,594

* PRV-positive test results ^#^ PRV-positive test results (cases) plus negative test results.

**Table 3 pathogens-09-00276-t003:** Estimated overall PRV seroprevalence in wild boar in Germany.

				95% CI	
Data Set	Time Interval	Sample Size	Prevalence (%)	LCL ^1^	UCL ^1^
Original	Total	149,594	13.73	13.55	13.90
	1	74,797	12.91	12.67	13.15
	2	74,797	14.54	14.29	14.80
Adjusted	1	74,797	8.71	--	--
	2	74,797	13.40	--	--

CI: confidence interval; ^1^ LCL: lower confidence limit; UCL: upper confidence limit.

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
