# Peer review of "Monitoring of Pseudorabies in Wild Boar of Germany—A Spatiotemporal Analysis"

_pathogens, 2020, doi:10.3390/pathogens9040276_

Round 1

Reviewer 1 Report

The submitted article has excellent quality. The methodology is well designed, the results are well analyzed, discussed and illustrated. The article offers very relevant information for the scientific community. I have just minor considerations:

Could the authors provide/discuss, if is it possible, information about:

  • the distance of Pseudorabies seroprevalence in wild boar from the initial analyzed year point;
  • the association between locations abundance of wild boar and PRB seroprevalence detection,
  • the orientation and shape of a distribution and dispersion- directional trends and dispersion- of PRB prevalence.

Line 326- “However, phylogenetic evidence suggests at least two different PRV variants and circulating in wild boar populations in Germany”. The information is correct, but the previous analyzed samples were from 1985-2008. Are there any phylogenetic analyses in the samples from the other years to confirm the pattern of variants in Germany?

References- in bibliographic references there are small errors (bold, commas, some references have doi and others do not). I kindly ask you to check it again.

Author Response

Please, see Attachement.

Reviewer 2 Report

The manuscript "Monitoring of Pseudorabies in Wild Boar of Germany – a spatiotemporal Analysis” by Denzin et. al. presents interesting results of a nationwide serological investigation of PRV infection.

This monitoring campaign collected data on more than 100,000 serum samples, which is an impressive number knowing the voluntary nature of participation. Analysis performed by the authors showed a PRV seroprevalence ranging between 0.14% and 31.4% depending on the federal state. Authors identified a large area in Eastern Germany and other smaller locations where the relative risk of a positive PRV result is high. This study brings new elements on the PRV distribution on the German territory but in addition, authors give recommendations for effective monitoring of PRV infection in wild boar, which is of particular interest to people working on this topic.

The article is well written, the analyses are sound, the data clearly presented and the results are well discussed.

Please find below some minor remarks/suggestions.

1/ the Materials and Methods section is placed before the Results part whereas according to the Instructions for Authors, it should be after the Discussion section.

2/ lines 168-175: are all areas with a high density of wild boar represented?In my opinion, it would be useful to have data on boar densities and their distribution in the territory from the beginning of the manuscript. A map presenting the average wild boar hunting bag (Figure 5) is only given in the discussion, what is too late.

3/ Table 1: the symbol “(“ in line “RP” can be removed

4/ Figures 2 and 3: The legends integrated in the figure are difficult to read because they are too small.

5/ Discussion: it is not necessary to indicate “Table XX” or “Figure XX” in this section.

6/ line 358: AS? Please, correct.
